# A comparison of CXR-CAD software to radiologists in identifying COVID-19 in individuals evaluated for Sars CoV-2 infection in Malawi and Zambia

**Sam Linsen**[1], **Aurélie Kamoun**[1], **Andrews Gunda**[2], **Tamara Mwenifumbo**[2], **Chancy Chavula**[2], **Lindiwe Nchimunya**[2], **Yucheng Tsai**[2], **Namwaka Mulenga**[2], **Godfrey Kadewele**[3], **Eunice Nahache Kajombo**[3], **Veronica Sunkutu**[4], **Jane Shawa**[5], **Rigveda Kadam**[1], **Matthew Arentz**[1] *

**1** FIND, Geneva, Switzerland, **2** Clinton Health Access Initiative, Lusaka, Zambia, **3** Health Technical Support Services, Malawi Ministry of Health, Lilongwe, Malawi, **4** Radiology Department, University Teaching Hospital, Zambia Ministry of Health, Lusaka, Zambia, **5** Radiology Department, Levy Mwanawasa Medical University, Zambia Ministry of Health, Lusaka, Zambia

* Matthew.arentz@me.com

**Data Availability Statement:** All data used in this submission can be accessed by emailing a request to FIND (info@finddx.org) or through the contact

## Abstract

AI based software, including computer aided detection software for chest radiographs (CXR-CAD), was developed during the pandemic to improve COVID-19 case finding and triage. In high burden TB countries, the use of highly portable CXR and computer aided detection software has been adopted more broadly to improve the screening and triage of individuals for TB, but there is little evidence in these settings regarding COVID-19 CAD performance. We performed a multicenter, retrospective cross-over study evaluating CXRs from individuals at risk for COVID-19. We evaluated performance of CAD software and radiologists in comparison to COVID-19 laboratory results in 671 individuals evaluated for COVID-19 at sites in Zambia and Malawi between January 2021 and June 2022. All CXRs were interpreted by an expert radiologist and two commercially available COVID-19 CXR-CAD software. Radiologists interpreted CXRs for COVID-19 with a sensitivity of 73% (95% CI: 69%- 76%) and specificity of 49% (95% CI: 40%-58%). One CAD software (CAD2) showed performance in diagnosing COVID-19 that was comparable to that of radiologists, (AUC-ROC of 0.70 (95% CI: 0.65–0.75)), while a second (CAD1) showed inferior performance (AUC-ROC of 0.57 (95% CI: 0.52–0.63)). Agreement between CAD software and radiologists was moderate for diagnosing COVID-19, and agreement was very good in differentiating normal and abnormal CXRs in this high prevalent population. The study highlights the potential of CXR-CAD as a tool to support effective triage of individuals in Malawi and Zambia during the pandemic, particularly for distinguishing normal from abnormal CXRs. These findings suggest that while current AI-based diagnostics like CXR-CAD show promise, their effectiveness varies significantly. In order to better prepare for future pandemics, there is a need for representative training data to optimize performance in key populations, and ongoing data collection to maintain diagnostic accuracy, especially as new disease strains emerge.

form at https://www.finddx.org/contact-us/.
However, all patient medical images are subject to
Malawi and Zambia data privacy laws and may not
be freely available. De-identified CAD software
results and metadata can be provided upon
request, while anonymized images may be shared
pending approval from the Malawi and Zambia
Ministries of Health (MoH). Requests for images
can be directed to the Ministry of Health (https://
www.moh.gov.zm/, eresconvergeltd@gmail.com)
in Zambia; and the Ministry of Health (https://www.
health.gov.mw, research@mail.gov.mw) in Malawi.

**Funding:** This work, including the data collection,
protocol development, analysis, and publication
was supported with funding provided by German
Ministry for Education and Research (BMBF)
through KfW 2020 62 156. The funders had no role
in study design, data collection and analysis,
decision to publish, or preparation of the
manuscript.

**Competing interests:** The authors have declared
that no competing interests exist.

## Author summary

During the COVID-19 pandemic, AI-based software was developed to help identify and
manage cases, including computer aided detection software to interpret chest X-rays
(CXRCAD). This technology has also been used in high tuberculosis (TB) burden coun-
tries to screen and manage TB cases. However, there's limited information on how well
these tools work for COVID-19 and other CXR features in these settings. We examined
chest X-rays from people at risk for COVID-19 in Zambia and Malawi to evaluate perfor-
mance of CXR-CAD software against expert radiologists and laboratory COVID-19 tests.
The research included X-rays from 671 participants, reviewed by two AI software pro-
grams and radiologists. The results showed that radiologists had a sensitivity of 73% and
specificity of 49% in detecting COVID-19. One AI software (CAD2) performed similarly
to radiologists, while another (CAD1) performed worse. The agreement between the AI
software and radiologists varied, but both were good at distinguishing between normal
and abnormal X-rays. The study suggests that while AI tools like CXR-CAD show poten-
tial, their effectiveness can vary. To improve these tools for future pandemics, more repre-
sentative training data and continuous data collection are necessary.

## Introduction

The COVID-19 pandemic, caused by the coronavirus SARS CoV-2, has had devastating conse-
quences for healthcare systems worldwide. By the end of 2023, there were over 7.7 million
reported deaths, and over 18 million estimated deaths globally [1]. A breakdown in global
coordination regarding testing, vaccination, and allocation of resources has been described as
a major factor involved in the failure of this global response [2]. Limited access to adequate
testing and treatment during the pandemic was a consequence of this breakdown, and likely
contributed to deaths due to COVID-19 and a number of other communicable diseases [3].

Use of medical imaging to help diagnose COVID-19 and differentiate it from other respira-
tory conditions remains a challenge given the overlapping clinical and radiological manifesta-
tions of respiratory illnesses. Chest X-rays (CXRs) have been a valuable tool in the evaluation
and diagnosis of respiratory diseases for decades. While not as sensitive as computed tomogra-
phy (CT) scans, CXRs are more widely available, less costly, less infrastructure intensive and
can be performed with minimal exposure risk to healthcare providers. Newer, highly portable
digital CXR devices have been deployed in many high burden tuberculosis (TB) settings,
improving access to medical imaging outside of regional hospitals, and CXRs may aid in the
identification of individuals with communicable respiratory diseases including those at greater
risk for worsening [4–6]. Such interventions may also be beneficial when future respiratory
pandemics arise. Yet, the interpretation of CXR findings, especially in a setting where the case-
load of patients with respiratory symptoms is high, requires novel strategies given variation in
access to radiologists and expert readers [7].

Leveraging artificial intelligence (AI) to interpret CXRs (computer aided detection, or
CXR-CAD) showed promise early in the COVID-19 pandemic. In research settings, use of
COVID-19 algorithms with CXR-CAD reported very high performance. CXR-CAD software
have been deployed in many high burden TB settings and are increasingly available for use in
global populations at risk for TB. Additionally, many CXR-CAD developers added COVID-19
specific algorithms during the pandemic and have other algorithms for CXR features which
are in use [8–10]. In many instances, training data for these software leveraged datasets drawn

from populations outside of the African continent, and it has been unclear how COVID-19 specific CAD algorithms could perform in populations in the region.

In addition to uncertainty on performance, as the pandemic evolved, policy shifted away from medical imaging (and CXR) as a first step in the diagnostics evaluation of an individual at risk for COVID-19. This has limited the understanding of the benefit of CXR-CAD for this use [11]. As a consequence of a decrease in reimbursement for COVID-19 specific screening and diagnostic tools, many commercially available COVID-19 CAD algorithms have been removed from the market, while the potential benefit for use has never been fully defined.

Understanding how AI based algorithms for COVID-19 that interpret CXRs can benefit at risk populations in Africa can better inform their potential for use in this region for future respiratory pandemics, especially in settings where access to radiologists may be limited. This is especially true if future respiratory pandemics impact populations at risk for TB where such software may be used to evaluate for non TB CXR features, and can be immediately available for use. Malawi and Zambia are two African countries with innovative digital health strategies which consider governance of digital technology to ensure local benefit [12,13]. In this study, we explore the potential benefits, challenges, and applications of CXR-CAD systems for use in diagnosing COVID-19 and CXR abnormalities in populations at risk for COVID-19 from Zambia and Malawi. In doing so, we aim to elucidate the role that AI specific diagnostics for CXR could play in the diagnosis and disease management COVID-19 and future respiratory pandemics.

## Materials and methods

### Study design and participants

We performed a multicenter, retrospective cross-over study evaluating CXRs from individuals at risk for COVID-19 with both CXR-CAD software and radiologist in comparison to WHO approved COVID-19 laboratory testing. Individuals were enrolled from one of two sites in Zambia (the University Teaching Hospital and the Levey Mwanawasa University Teaching Hospital, both in Lusaka, Zambia) and one of five sites in Malawi (Mzuzu Central Hospital, Kamuzu Central Hospital, Queen Elizabeth Central Hospital, Nsanje District Hospital, and Mwaiwathu Hospital). Included patients were adults evaluated at the enrolling sites from January 1, 2021 to June 1, 2022 who had a presentation consistent with COVID-19 based on a clinical evaluation; and who also had received both a WHO approved Sars CoV-2 test as well as a digital chest radiograph within 72 hours of evaluation. Individuals younger than 18 years of age, and those who did not have a result for COVID-19 testing and a CXR digital image accessible in Digital Imaging and Communications in Medicine (DICOM) format were excluded.

### Sample size and sampling

For the comparison of sensitivity and specificity between radiologist and CXR-CAD diagnostics, a sample size calculation was generated using a previously established formula for comparison of proportions in cross-over designs [14]. The sample size of 500 total participants per country was calculated based on existing data to demonstrate a difference in accuracy of 10% between CXR- CAD systems and human readers at a COVID-19 prevalence of 20%, and was powered to determine a sensitivity of 90% and a specificity of 60% against laboratory reference standards at a COVID prevalence of 20%.

### Data collection

Records from all patients with COVID-19 testing during the study period were reviewed consecutively by researchers at each site to determine eligibility. In individuals meeting inclusion

criteria, data were collected and anonymized for evaluation using Open Clinica (Waltham, USA). Additionally, DICOM CXR images from included individuals were obtained by site researchers. Prior to sharing of DICOM images with FIND, identifying information was removed by use of a previously described DICOM anonymizing tool [15]. Patient clinical data and radiologist interpretations were aggregated and duplicate study subjects were excluded prior to analysis.

## Reference standards

The COVID-19 laboratory reference standard was defined based on the result of a WHO-approved Sars CoV-2 diagnostic test [16]. COVID-19 cases were defined as positive by one or more diagnostic test results within 72 hours of clinical evaluation. COVID-19 controls were defined as negative by all diagnostic test results performed within 72 hours of clinical evaluation.

Our radiology reference standard (RRS) for COVID-19 drew on expert radiologists (> 5 years experience) from Malawi and Zambia who were blinded to COVID-19 testing results. A single radiologist from the country where the image was acquired independently evaluated CXRs for findings and recorded results as described below. Radiologists were encouraged to use published guidelines on chest radiograph interpretation for COVID-19 as a component of their interpretation [17].

Expert radiologists characterized CXRs in one of the following 3 categories:

1. CXR pattern consistent with COVID-19

2. CXR pattern abnormal, but findings not consistent with COVID-19

3. CXR pattern normal

For the primary analysis, RRS were considered positive for COVID-19 if radiologist interpretation determined the CXR pattern was consistent with COVID-19 (category #1). The RRS was considered negative for COVID-19 if the interpretation determined the CXR pattern was normal, or was abnormal, but with findings not consistent with COVID-19 (categories #2 and #3).

For the secondary analysis, radiologist evaluations were considered abnormal if interpretation of the CXR was characterized as abnormal and consistent with COVID-19, or abnormal, but with findings non consistent with COVID-19 (categories #1 and #2). Radiologist evaluations were considered normal only for CXRs interpreted with findings consistent with a normal CXR pattern (category #3).

In instances where multiple DICOM CXR images were available within 72 hours of evaluation, the first CXR chronologically was collected for analysis. In instances where there were duplicate DICOM images, the higher resolution/larger file was selected and used for evaluation. If these parameters were identical, one image was selected for analysis and any duplicates were deleted.

## Ethics and privacy statement

The study received approval through the Clinton Health Access Initiative Institutional Review Board (CHAI IRB), and received in country ethical/IRB approval. Because of this study's retrospective design, informed consent was not obtainable. In all instances data was anonymized on site prior to sharing with FIND for evaluations. Data and images were stored in an encrypted password protected storage, with full data only available to the study PI and relevant members of the FIND data science team.

### Change in approach to developer evaluations during the study

Initially, a comparative analysis of multiple COVID-19 CAD algorithms was planned, including multiple developers FIND previously identified with CXR-CAD algorithms for TB [9,18]. However, as the pandemic evolved, many CXR-CAD developers removed their COVID 19 algorithms from commercial use. As a result, FIND established an agreement with 2 CXR-CAD developers to independently evaluate performance of their COVID-19 CXR-CAD and normal/abnormal CXR-CAD algorithms that had been in commercial use during the height of the pandemic, in this cohort of individuals at risk for COVID-19. However, given the removal of many of these products from the market, a stipulation to this evaluation was that publicly available results would blind specific developer performance in this analysis. Both developers included in this analysis have CXR-CAD products in use for the evaluation of CXRs for TB, and both have received WHO stringent regulatory approval for at least one diagnostic use for CXR- CAD. For this publication, these software developers are referred to as CAD1 and CAD2.

### Data analysis

For the primary analysis, both radiologist and COVID-19 algorithms were evaluated in comparison to a WHO approved laboratory reference standard. For CXR-CAD software interpretation, images were first pre-processed as needed to conform to CXR-CAD software developer specifications for DICOM files. Images were then exposed to CXR-CAD software with outputs/probability scores recorded for COVID-19 algorithms and for normal/abnormal algorithms (i.e. determining if any abnormalities are observed), using the FIND validation platform, as has been described elsewhere [19]. Performance metrics (sensitivity and specificity) of CXR-CAD for COVID-19 were compared to radiologist readings using CXR-CAD thresholds set to observed radiologist sensitivity and, separately, to observed radiologist specificity. This analysis mirrors a similar approach that has been described in other studies evaluating CXR-CAD for TB [15,20]. Subgroup analysis was performed based on country, site, age group, sex, symptoms, diabetes status (if known), and HIV status (if known). Results were presented for CXR-CAD sensitivity and specificity with 95% confidence intervals.

CXR-CAD scores were used to generate receiver operating characteristic (ROC) curves for both COVID scores and any abnormalities scores [21]. The Area Under the ROC curve (AUC-ROC) for each CAD system was calculated against laboratory reference standards using binomial distribution assumptions for the primary analysis.

Radiologist sensitivity and specificity assessments for COVID-19 were calculated in comparison to laboratory reference standards. CXR-CAD software estimates of sensitivity and specificity were then calculated at the threshold produced by the same sensitivity or specificity achieved by the radiologist.

For the secondary analysis, CXR-CAD was evaluated for agreement with a radiologist in differentiating normal and abnormal CXRs, based on the AC1 coefficient [22]. In this secondary analysis, CXR-CAD algorithms for abnormal CXRs were evaluated using the manufacturer suggested threshold.

## Results

### Population characteristics and image assessment

In total 758 CXR images were available for this study. A total of 749 CXR images were read with findings reported by radiologist and 9 were not. Among the shared reads, 13 CXR images had to be excluded from this study as they were not properly de-identified; for 11 patients, age

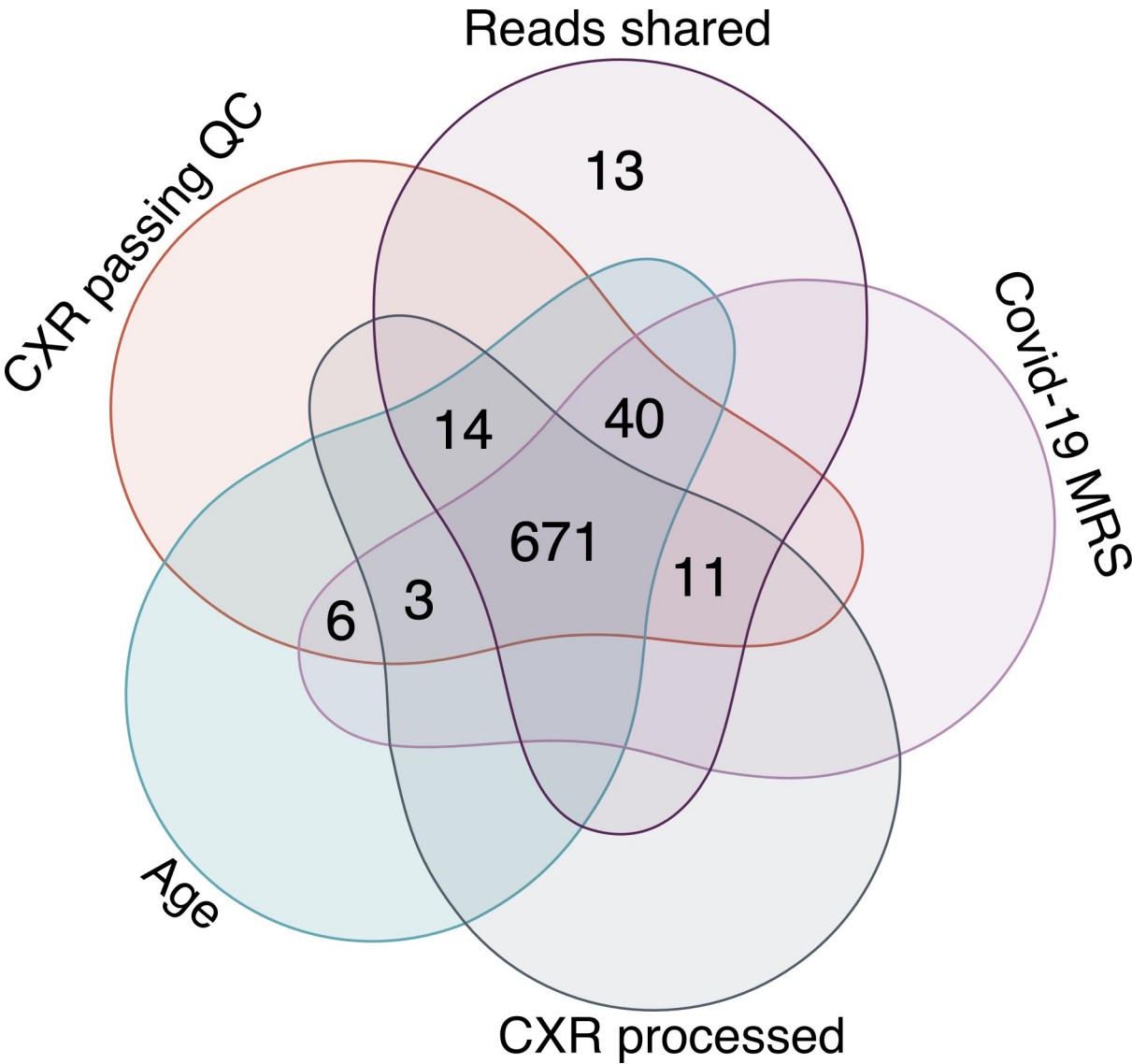

**Fig 1.** The distribution of participants in this study based on the available data: participants were selected based on Reads shared (CXR interpretation by a radiologist); available MRS diagnostic output (Covid-19 MRS); CXRs processed by CAD-1 and CAD-2 (CXR processed); and quality checks (CXR passing QC).

information was missing and these files were excluded; and 14 patient entries did not report a COVID-19 test which was recorded withing 72 hours of evaluation. In the remaining 711 images, there were 40 images which could not be processed by CAD1 or CAD2, and the associated subjects were removed from the evaluation.

This left a total of 671 participant with complete clinical data and appropriate images available for evaluation (89% of the image total) which were included in the comparative covid analysis. A Venn diagram of participants inclusion in the final analyses based on image assessment and processing is shown in **Fig 1**.

The included participants included 269 (40% of the total) from Malawi and 402 (60% of the total) from Zambia. This total included 540 COVID-19 cases and 131 controls, with a considerably higher prevalence of COVID-19 cases than in our pre-study power calculations. Of the

total number of cases, 212 (39%) were from Malawi and 328 (61%) were from Zambia. Of the total controls, 57 (44%) were from Malawi and 74 (56%) were from Zambia.

The majority of participants were male (n = 410, 61%) and the average age was 51 years (range 18 to 90 years). In most instances, the test sample was collected via nasopharyngeal swab (N = 474, 71%). A minority of patients were tested HIV positive (n = 77, 11%) or were known to be diabetic (n = 86, 13%). A number of individuals were listed as having absent symptoms or symptoms were not recorded. However, of those that were Sars CoV-2 positive, 18 of 22 individuals with absent symptoms, and 33 of 39 individuals with no symptoms recorded were given a clinical diagnosis of pneumonia. It is unclear how uncaptured data (for instance, clinical exam findings) contributed to this discrepancy. A description of the populations by site are described in **Table 1**.

## Radiologist Reference Standard (RRS) for COVID and radiologist interpretation other abnormalities

Radiologists interpreted 460 images (72.8% of cases and 51.2% of controls, based on microbiologic reference standard or MRS) as having findings consistent with COVID-19. Of the remaining COVID cases positive by MRS, radiologists interpreted the majority (89, 16.5%) as abnormal but with findings not consistent with COVID-19 pneumonia. Details of radiologist interpretation of CXRs for COVID-19 are shown **in Table 2**.

In comparison to the COVID-19 laboratory testing (MRS), blinded expert radiologists had a pooled specificity of 49% (95% CI 40% - 58%) and a pooled sensitivity of 73% (95% CI 69%-76%) for diagnosis of COVID-19. However, specificity varied significantly by country. In Zambia, the observed specificity was 30% (95% CI 20%-41%) and sensitivity was 73% (95% CI 68%-78%), while in Malawi, radiologist observed specificity was 74% (95% CI 60%-84%) and sensitivity was 73% (95% CI 66%-79%) for COVID-19. Radiologist diagnostic accuracy is shown in **Table 3**, **Figs 2 and 3**.

## Software interpretation of CXR images for COVID-19

Two commercially available CAD software were included in this analysis (deemed CAD1 and CAD2). CAD software interpretation of CXRs was compared to COVID-19 laboratory testing and radiologist performance. The diagnostic outcome of these 2 CAD software is reflected by a score that is generated from a CXR image, and then defined as positive or negative according to a set threshold.

The observed area under the receiver operating curve (AUC-ROC) for CAD1 was 0.57 (95% CI 0.52–0.63) and for CAD2 was 0.70 (95% CI 0.65–0.75). Pooled estimates for CAD1 had observed 95% confidence intervals that were lower that our RRS for COVID-19 and without overlapping 95% CI, when software thresholds were set at the radiologist observed sensitivity and specificity (**Table 4**). CAD2 had an observed software performance that was comparable to the observed radiologist performance.

Although CAD1 had a similar performance in Malawi (AUC 0.55, 95% CI 0.47, 0.63) and in Zambia (AUC 0.56, 95% CI 0.49, 0.64), given the lower observed specificity in Zambia, performance of CAD1 was comparable to the RRS, with overlapping 95% CIs in the Zambia population (**Fig 3**). CAD2 demonstrated an AUC of 0.75 (95% CI of 0.68, 0.82) in Malawi and 0.66 (95% CI 0.59, 0.72) in Zambia had an observed performance comparable to a RRS at set specificities in CXRs from both countries.

We evaluated the agreement between CAD software and radiologists at vendor recommended thresholds, and found that agreement was very low for CAD1, with only 60%

**Table 1. Demographic and clinical characteristics from included participants PCR: Sars CoV-2 polymerase chain reaction testing, Antigen: Sars CoV-2 Antigen test.**

| Variable | Total | Positive Cases | Negative Cases |
|---|---|---|---|
| **All** | **671** | **540** | **131** |
| **Study Site ID** | | | |
| **Zambia** | | | |
| UTH | 190 | 142 (26.3%) | 48 (36.6%) |
| LEVY M. UTH | 212 | 186 (34.4%) | 26 (19.8%) |
| **Malawi** | | | |
| Mzuzu Central Hospital | 30 | 30 (5.6%) | 0 (0.0%) |
| Kamuzu Central Hospital | 105 | 101 (18.7%) | 4 (3.1%) |
| Queen Elizabeth Central Hospital | 62 | 62 (11.5%) | 0 (0.0%) |
| Nsanje District Hospital | 8 | 8 (1.5%) | 0 (0.0%) |
| Mwaiwathu Hospital | 64 | 11 (2.0%) | 53 (40.5%) |
| **Age group** | | | |
| [18–25] | 36 | 30 (5.6%) | 6 (4.6%) |
| [26–35] | 87 | 60 (11.1%) | 27 (20.6%) |
| [36–45] | 141 | 111 (20.6%) | 30 (22.9%) |
| [46–55] | 145 | 118 (21.9%) | 27 (20.6%) |
| [56–65] | 128 | 107 (19.8%) | 21 (16.0%) |
| [> = 66] | 134 | 114 (21.1%) | 20 (15.3%) |
| **Gender** | | | |
| Female | 260 | 203 (37.6%) | 57 (43.5%) |
| Male | 410 | 337 (62.4%) | 73 (55.7%) |
| Not recorded | 1 | 0 (0.0%) | 1 (0.8%) |
| **Type of COVID-19 test done** | | | |
| Antigen | 154 | 90 (16.7%) | 64 (48.9%) |
| PCR | 516 | 449 (83.1%) | 67 (51.1%) |
| *NA* | *1* | *1 (0.2%)* | *0 (0.0%)* |
| **HIV status** | | | |
| Positive | 77 | 71 (13.1%) | 6 (4.6%) |
| Negative | 459 | 381 (70.6%) | 78 (59.5%) |
| Not recorded | 135 | 88 (16.3%) | 47 (35.9%) |
| **Diabetes status** | | | |
| Positive | 86 | 70 (13.0%) | 16 (12.2%) |
| Negative | 450 | 382 (70.7%) | 68 (51.9%) |
| Not recorded | 135 | 88 (16.3%) | 47 (35.9%) |
| **Symptoms** | | | |
| Present | 586 | 479 (88.7%) | 107 (81.7%) |
| Absent | 35 | 22 (4.1%) | 13 (9.9%) |
| Not recorded | 50 | 39 (7.2%) | 11 (8.4%) |
| **Sampling method** | | | |
| Bronch-alveolar lavage | 1 | 1 (0.2%) | 0 (0.0%) |
| Nasal swab | 186 | 173 (32.0%) | 13 (9.9%) |
| Nasopharyngeal swab | 474 | 356 (65.9%) | 118 (90.1%) |
| Not recorded | 3 | 3 (0.6%) | 0 (0.0%) |
| Throat swab | 7 | 7 (1.3%) | 0 (0.0%) |

**Table 2. Radiologist interpretation of CXRs in comparison to laboratory testing results.** Positive and Negative cases are defined based on the Sars CoV–2 Microbiologic Reference Standard (MRS).

| Human interpretation of CXR | | Total | MRS Positive | MRS Negative |
|---|---|---|---|---|
| *COVID-19 Positive* | CXR pattern consistent with COVID-19 | 460 | 393 (72.8%) | 67 (51.2%) |
| *COVID-19 Negative* | CXR pattern abnormal, but findings not consistent with COVID-19 (non COVID-19) | 116 | 89 (16.5%) | 27 (20.6%) |
| | CXR pattern normal (no abnormalities) | 95 | 58 (10.7%) | 37 (28.2%) |

agreement regarding COVID-19 cases and controls, having an AC1 coefficient value of 0.28. Agreement was moderate for CAD2 (75%) with a AC1 of 0.53. (S1 and S2 Tables)

Given the poor observed performance, we assessed whether a difference in COVID-19 strains could have resulted in a difference in algorithm performance [23]. Therefore, we performed an analysis evaluating performance before and after November of 2021 (when the Omicron strain first superseded the Delta strain as the dominant variant in South Africa, a country close geographically to Zambia and Malawi) to explore for a difference in software performance, in comparison to a radiologist. As shown in **Fig 4**, The performance of CAD1 demonstrated an AUC of 0.55 (95% CI 0.48, 0.61) before the emergence of the Omicron strain and an AUC of 0.64 (95% CI 0.53,0.75) after the emergence of Omicron. CAD2 had an observed AUC of 0.71 (95% CI 0.65–0.77) before the emergence of Omicron, and an observed AUC of 0.65 (95% CI 0.55–0.75) after this time for COVID-19 diagnosis. Radiologist sensitivity also decreased after emergence of the Omicron strain, and observed CAD2 software performance remained comparable to a radiologist as set sensitivity and specificity with overlapping 95% confidence margins in both subgroups.

Additionally, we postulated that software performance may have differed depending on the Sars CoV-2 laboratory reference standard used. When separating antigen and PCR testing results, we found that the observed specificity of the blinded expert radiologist and the AUC of CAD 1 both were higher in the subgroup evaluated for COVID-19 with antigen testing (Fig 5).

## Software analysis by subgroup

We also evaluated subgroups by age, gender, HIV status, and diabetes status. Given the low numbers in each of these groups, we observed large confidence intervals in AUCs with non significant trends. Further subgroup analysis is shown in the supplemental material (S1 to S4 Figs).

## Software interpretation of normal vs. abnormal CXR images

For our secondary aim, we evaluated the agreement between CAD software and radiologists at identifying any abnormalities on CXRs. Of the includes CXRs, two lacked a radiologist interpretation of normal vs. abnormal; and 5 additional CXR images read by CAD2 which produced a COVID-19 score did not produce a normal/abnormal score. Therefore, the total number of images included in this secondary analysis were 669 for CAD1 and 664 for CAD2.

**Table 3. Radiologist accuracy in identifying COVID-19 on CXR in comparison to laboratory test results.** N: number included, RRS: Radiologist Reference Standard, CI: confidence interval.

| | N | RRS Pos | RRS Neg | Sensitivity (95% CI) | Specificity (95% CI) |
|---|---|---|---|---|---|
| *Overall* | 671 | 460 | 211 | **73% (69–76%)** | **49% (40–58%)** |
| *Zambia* | 402 | 291 | 111 | **73% (68–78%)** | **30% (20–41%)** |
| *Malawi* | 269 | 169 | 100 | **73% (66–79%)** | **74% (60–84%)** |

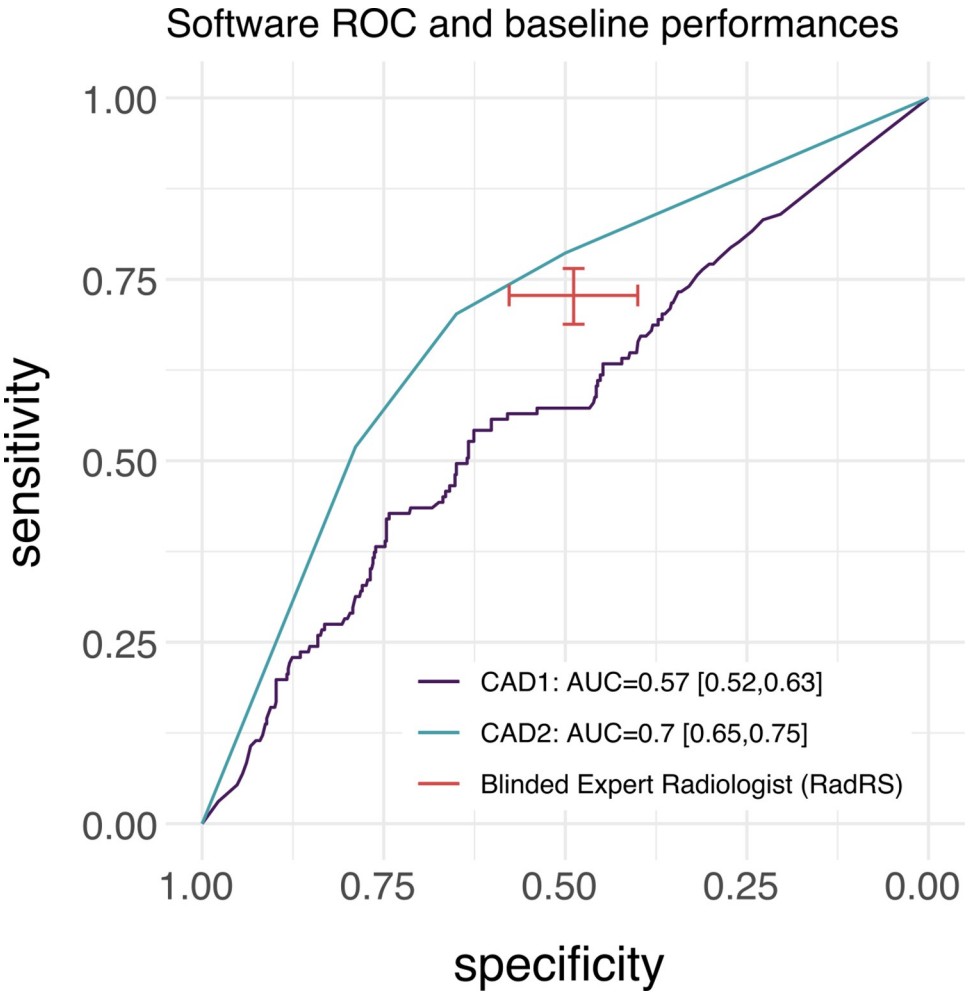

**Fig 2. Aggregate performance and 95% confidence intervals of radiologist and software CAD1 and CAD2 at identifying Covid-19 in comparison to baseline molecular testing. AUC: Area under Curve, RadRS: Radiologist reference standard for COVID-19.** CAD1: Computer Aided Detection software 1, CAD2: Computer Aided Detection software 2.

For this secondary aim, agreement of both CAD software with a radiologist was very good, with 89% agreement and a calculated agreement coefficient of 0.86 (Table 5B). In settings where a radiologist identified a CXR as "abnormal" level of agreement with software was better, with 93% of images read by CAD1 and 95% of images read by CAD2 as also identifying CXRs as abnormal. (Table 5A)

## Discussion

In this study, we independently evaluated performance of CXR-CAD software, in comparison to a radiologist, at identifying COVID-19 or other radiographic abnormalities in individuals evaluated for COVID-19 in two countries in Africa. In doing so, we aim to highlight the potential uses and benefits of CXR-CAD software for COVID-19 and consider the implications for use in future respiratory pandemics. There are a number of key takeaways from this work.

First, we found that performance of CXR-CAD in evaluating CXRs for findings consistent with COVID-19 varied for the two software's evaluated. In comparison to a radiologist against

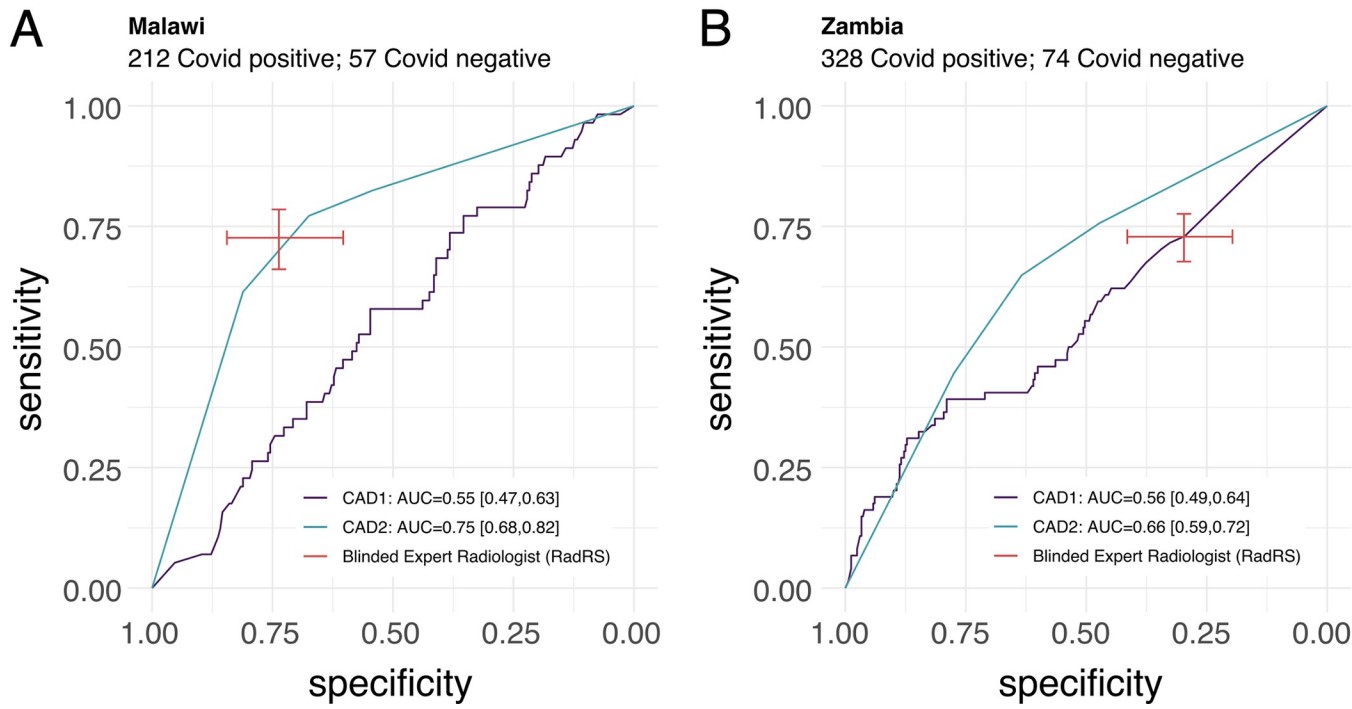

**Fig 3. Performance of radiologist and software CAD1 and CAD2 at identifying Covid-19 in comparison to baseline molecular testing.** A Performance in the Malawi cohort; B performance in the Zambia cohort. AUC: Area under Curve, RadRS: Radiologist reference standard for COVID-19. CAD1: Computer Aided Detection software 1, CAD2: Computer Aided Detection software 2.

pooled data from Zambia and Malawi, one software performance (CAD1) was inferior to an expert human radiologist in diagnosing COVID-19, but a second (CAD2) was comparable. It is unclear how different the training data sets were for these two software, or how well they aligned with the populations we evaluated these tools in. As is the case for many commercial products, details related to algorithm development and training are not publicly available. However, these key aspects in software training data are known to be central to appropriate development and deployment of AI based tools [24,25]. Such issues are critical to address during a rapidly evolving respiratory pandemic, where digital data are limited, and where clinical presentation and radiographic manifestations may change. Our findings suggest that use of CXR-CAD to identify novel respiratory infections can achieve performance comparable to human interpretation. However, it is likely that development of CXR-CAD for future

**Table 4. Software performance at a set radiologist observed sensitivity and specificity.** Fixed values (threshold, sensitivity, specificity) were mapped to their closest point from the ROC. This may lead to slight differences between the actual fixed value and the corresponding value from the ROC. Th: threshold Sens: Sensitivity Spec: Specificity CI: confidence interval CAD1: Computer Aided Detection software 1, CAD2: Computer Aided Detection software 2.

| Description | Fixed Value | CAD1 | | | CAD2 | | |
|---|---|---|---|---|---|---|---|
| | | Th | Sens (95% CI) | Spec (95% CI) | Th | Sens (95% CI) | Spec (95% CI) |
| **Fixed Threshold** | | | | | | | |
| Vendor recommended | Th | 0.55 | 47% [38%-55%] | 66% [62%-70%] | 1.5 | 70% [62%-78%] | 65% [61%-69%] |
| **Fixed Sensitivity** | | | | | | | |
| Radiologist performance | Sens | 0.99 | 73% [66%-80%] | 35% [31%-39%] | 1.5 | 70% [62%-78%] | 65% [61%-69%] |
| **Fixed Specificity** | | | | | | | |
| Radiologist performance | Spec | 0.91 | 61% [53%-69%] | 46% [42%-50%] | 2.5 | 79% [71%-85%] | 50% [46%-54%] |

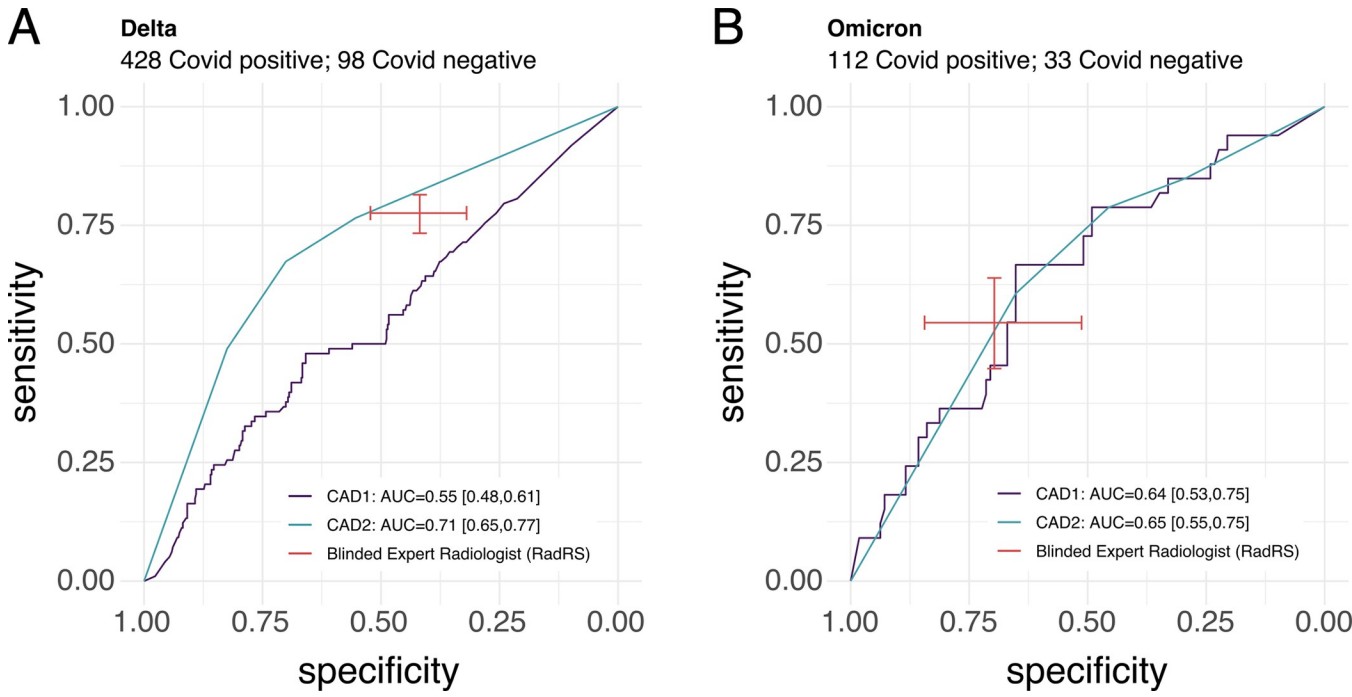

**Fig 4. Software and radiologist performance at diagnosing COVID-19 before and after the shift to Omicron being the dominant variant.** Delta: results coinciding with participants evaluated for COVID-19 before November 1, 2021 (A) and, Omicron: results coinciding with participants evaluated for COVID-19 after November 1, 2021 (B) AUC: Area under Curve, RadRS: Radiologist reference standard for COVID-19. CAD1: Computer Aided Detection software 1, CAD2: Computer Aided Detection software 2.

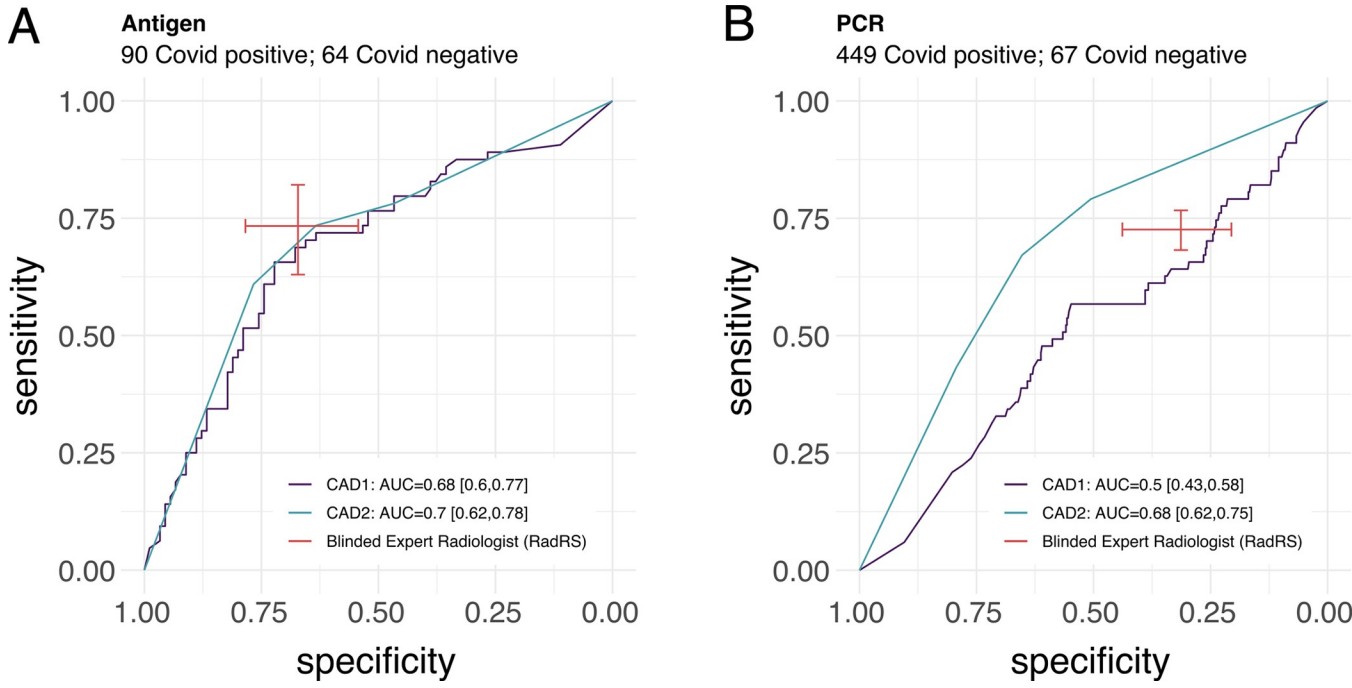

**Fig 5.** Software and radiologist performance at diagnosing COVID-19 in subgroups evaluated with antigen (A) or PCR (B) testing for SARS CoV-2. AUC: Area under Curve, RadRS: Radiologist reference standard for COVID-19. CAD1: Computer Aided Detection software 1, CAD2: Computer Aided Detection software 2, PCR: Sars CoV-2 polymerase chain reaction testing, Antigen: Sars CoV-2 Antigen test.

**Table 5. (A) Agreement of CAD software with radiologists, at manufacturer suggested thresholds for normal vs. abnormal CXRs.** CAD1: Computer Aided Detection software 1, CAD2: Computer Aided Detection software 2. Percentages group each radiologist assessment according to the corresponding CAD outcome. (B) Percentage agreement of CAD software with radiologists, at manufacturer suggested thresholds for normal vs. abnormal CXRs and the calculated agreement coefficient. N: number, AC1: Gwet's Agreement Coefficient, CAD1: Computer Aided Detection software 1, CAD2: Computer Aided Detection software 2.

| A | | | |
|---|---|---|---|
| Software | Software interpretation | Abnormal radiologist assessment | Normal radiologist assessment |
| CAD1 | Abnormal | 532 (93%) | 33 (35%) |
| CAD1 | Normal | 42 (7%) | 62 (65%) |
| CAD2 | Abnormal | 544 (95%) | 43 (47%) |
| CAD2 | Normal | 29 (5%) | 48 (53%) |
| B | | | |
| Software | N | Agreement | AC1 |
| CAD1 | 669 | 89.00% | 0.86 |
| CAD2 | 664 | 89.00% | 0.86 |

pandemics would require more training data from populations in Africa in order to optimize performance in these settings. However, if performance validation is possible, such tools may be considered as an alternative to a human radiologist, or to support through preliminary interpretations in settings in Zambia and Malawi where trained expert readers are scarce.

Second, we found that performance of CAD software and radiologist interpretation varied over time, and a decrease in sensitivity was observed which coincided with emergence of the Omicron wave of COVID-19. Coinciding with the time that Omicron superseded the Delta strain as the globally dominant variant, the rate of infections and clinical handling started to show a different dynamic: new strains were more infectious but may have resulted in decreased severity of disease, and immunization through prior exposure rates and vaccinations started to increase, triage practices and molecular test availability became more widespread. The generally milder symptomatic presentation may have resulted in a reduced specificity of radiologists; and a possible increase availability of antigen testing improved the benchmark for diagnosing acute disease and thereby improved the performance of CAD1. This finding highlights the need for novel strategies to re-train and fine tune algorithms when strains with different phenotypes and differing levels of respiratory involvement emerge [23]. In instances where AI based diagnostics are supporting clinical care, including those that use medical imaging, emergence of new strains should be a key consideration for software re-training and version updates.

Third, both software showed strong agreement with a radiologist in differentiating normal and abnormal CXRs in this cohort of individuals at risk for COVID-19. This was particularly true in settings where a radiologist identified no abnormalities on CXR. Algorithms for differentiating normal vs. abnormal CXRs may be an alternative for use early in a pandemic which can identify a subset of individuals with normal CXRs. As a result, when data is scarce and disease specific algorithms will take long to adequately train such models may be valuable in triaging those without respiratory involvement. While such a strategy would likely not have had a large influence on disease transmission, it could rapidly screen or triage a large group of *at risk* individuals to identify those who and do not need close monitoring or further testing, thereby optimize clinical resources and enhance robustness in the diagnostic network.

CXR-CAD software is a promising tool that is currently being used with portable CXR in many settings globally, and for diagnosis of TB and other radiographic findings. Some of these software were adapted for use in identifying COVID-19 during the height of the global pandemic. In the future, consideration of where and how these tools can be used may be a valuable consideration for future pandemics preparedness in Zambia and Malawi, and in other

countries in Africa as a component of a digital health strategy. However, as will most AI based diagnostics, key considerations need to be made regarding the collection and use of high quality, representative data for use in the training and testing of these products in the settings where they will be deployed, and in post deployment monitoring. This study demonstrates that some commercially available CAD software could achieve parity with a radiologist in performance during the pandemic, and that the key considerations mentioned above are central to maintaining benefit.

## Limitations

There are a number of limitations to this study. First, the retrospective nature and the inclusion criteria likely subset of individuals which may not be representative of a general population being triaged for COVID-19. A number of unadjusted factors associated with demographics, disease strains and phenotypes, and vaccination may have significantly impacted the results. Although radiologists were used as a primary comparator in a matched crossover evaluation, this limitation remains significant, and needs follow up in other use cases if and when such prospective data collection is possible. Second, there was a significant difference in radiologist performance between Malawi and Zambia. It is unclear whether this represented a difference in approach to CXR interpretation or other factors. Although having a country specific radiology standard has value, the difference in interpretation between different radiologist has bene extensively characterized in other literature and limits the generalizability. Third, Sars Co-V2 testing positivity via antigen test or PCR may have a significant impact on the clinical phenotype of COVID-19. Although we address this above, we could not control for this factor, given the retrospective nature of the study. Fourth, although a number of developers had developed algorithms for COVID-19 during the pandemic, many withdrew their software from the market and a comparison of more products could have added more insight into the findings. It is unclear how those other products would have performed against this dataset. Lastly, most CAD developers now offer some strategy for local threshold setting or fine tuning. Local fine tuning on a subset of images would often improves performance for a given setting and population. However, given the independent nature of our assessment, fine tuning on this dataset was not offered, but likely would have improved software performance.

## Supporting information

**S1 Fig. Subgroup analysis of performance of radiologist and software CAD1 and CAD2 at identifying COVID-19 in comparison to baseline molecular testing grouped by age.** Age 15–25 years (A), Age 26–35 years (B), age 36–45 years (C), age 46–55 years (D) age 56–65 years (E), and age 66 years or higher (F) AUC: Area under Curve, RadRS: Radiologist reference standard for COVID-19. CAD1: Computer Aided Detection software 1, CAD2: Computer Aided Detection software 2.
(TIF)

**S2 Fig. Subgroup analysis of performance of radiologist and software CAD1 and CAD2 at identifying COVID-19 in comparison to baseline molecular testing grouped by gender** Female (A) and male (B). AUC: Area under Curve, RadRS: Radiologist reference standard for COVID-19. CAD1: Computer Aided Detection software 1, CAD2: Computer Aided Detection software 2.
(TIF)

**S3 Fig. Subgroup analysis of performance of radiologist and software CAD1 and CAD2 at identifying COVID-19 in comparison to baseline molecular testing grouped by HIV status.**

HIV positive (A) and HIV negative (B) AUC: Area under Curve, RadRS: Radiologist reference standard for COVID-19. CAD1: Computer Aided Detection software 1, CAD2: Computer Aided Detection software 2.
(TIF)

**S4 Fig. Subgroup analysis of performance of radiologist and software CAD1 and CAD2 at identifying COVID-19 in comparison to baseline molecular testing grouped by diabetes status.** Diabetic (A) and non diabetic (B) AUC: Area under Curve, RadRS: Radiologist reference standard for COVID-19. CAD1: Computer Aided Detection software 1, CAD2: Computer Aided Detection software 2.
(TIF)

**S1 Table. Agreement of CAD software with radiologists, at manufacturer suggested thresholds for COVID-19 CAD1: Computer Aided Detection software 1, CAD2: Computer Aided Detection software 2.**
(DOCX)

**S2 Table. Percentage agreement of CAD software with radiologists, at manufacturer suggested thresholds for COVID-19 and the calculated agreement coefficient.** CAD1: Computer Aided Detection software 1, CAD2: Computer Aided Detection software 2 AC1: Gwet's Agreement Coefficient.
(DOCX)

## Acknowledgments

The authors would like to thank Nikhil Jagtiani and Stefano Ongarello at FIND for their support in the study design and execution. Lastly, we would like to thank Qure.ai and VinBrain for their support in this study.

## Author Contributions

**Conceptualization:** Aurélie Kamoun, Rigveda Kadam, Matthew Arentz.

**Data curation:** Sam Linsen, Aurélie Kamoun, Andrews Gunda, Tamara Mwenifumbo, Chancy Chavula, Lindiwe Nchimunya, Yucheng Tsai, Namwaka Mulenga, Godfrey Kadewele, Eunice Nahache Kajombo, Veronica Sunkutu, Jane Shawa, Rigveda Kadam, Matthew Arentz.

**Formal analysis:** Sam Linsen, Aurélie Kamoun, Matthew Arentz.

**Funding acquisition:** Rigveda Kadam, Matthew Arentz.

**Investigation:** Aurélie Kamoun, Andrews Gunda, Tamara Mwenifumbo, Chancy Chavula, Lindiwe Nchimunya, Yucheng Tsai, Namwaka Mulenga, Godfrey Kadewele, Eunice Nahache Kajombo, Veronica Sunkutu, Jane Shawa, Rigveda Kadam, Matthew Arentz.

**Methodology:** Aurélie Kamoun, Rigveda Kadam, Matthew Arentz.

**Project administration:** Rigveda Kadam, Matthew Arentz.

**Resources:** Rigveda Kadam, Matthew Arentz.

**Supervision:** Rigveda Kadam, Matthew Arentz.

**Validation:** Sam Linsen, Aurélie Kamoun, Rigveda Kadam, Matthew Arentz.

**Writing – original draft:** Sam Linsen, Aurélie Kamoun, Rigveda Kadam, Matthew Arentz.

**Writing – review & editing:** Sam Linsen, Andrews Gunda, Tamara Mwenifumbo, Chancy Chavula, Lindiwe Nchimunya, Yucheng Tsai, Namwaka Mulenga, Godfrey Kadewele, Eunice Nahache Kajombo, Veronica Sunkutu, Jane Shawa, Rigveda Kadam, Matthew Arentz.

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
