## [Decision Letter · Decision Letter 0]

9 Jul 2024

PDIG-D-24-00202

A Comparison of CXR-CAD Software to Radiologists in Identifying COVID-19 in Individuals Evaluated for Sars CoV-2 Infection in Malawi and Zambia

PLOS Digital Health

Dear Dr. Arentz,

Thank you for submitting your manuscript to PLOS Digital Health. After careful consideration, we feel that it has merit but does not fully meet PLOS Digital Health's publication criteria as it currently stands. Therefore, we invite you to submit a revised version of the manuscript that addresses the points raised during the review process.

Please submit your revised manuscript within 60 days Sep 07 2024 11:59PM. If you will need more time than this to complete your revisions, please reply to this message or contact the journal office at digitalhealth@plos.org. Please include the following items when submitting your revised manuscript:

We look forward to receiving your revised manuscript.

Kind regards,

Catherine G Bielick

Guest Editor

PLOS Digital Health

Journal Requirements:

Additional Editor Comments (if provided):

We appreciate your consideration of PLOS Digital Health for publication of your manuscript. It has been reviewed and it both reviewers are in agreement that there are many strengths and does fit within our mission and scope. The manuscript is suitable for revision and resubmission to reconsider for publication with the listed critique provided.

Reviewers' comments:

Reviewer's Responses to Questions

**Comments to the Author**

1. Does this manuscript meet PLOS Digital Health’s publication criteria? Is the manuscript technically sound, and do the data support the conclusions? The manuscript must describe methodologically and ethically rigorous research with conclusions that are appropriately drawn based on the data presented.

Reviewer #1: Partly

Reviewer #2: Yes

2. Has the statistical analysis been performed appropriately and rigorously?

Reviewer #1: No

Reviewer #2: Yes

3. Have the authors made all data underlying the findings in their manuscript fully available (please refer to the Data Availability Statement at the start of the manuscript PDF file)?

Reviewer #1: Yes

Reviewer #2: Yes

4. Is the manuscript presented in an intelligible fashion and written in standard English?

Reviewer #1: Yes

Reviewer #2: Yes

5. Review Comments to the Author

Reviewer #1: The authors performed a retrospective study to assess the performance of two software products in the interpretation of chest radiographs performed in Zambia and Malawi during the COVID-19 pandemic. The concordance of radiologist and software interpretations of CXR with respect to normality and COVID-19 specifically was assessed, as was the concordance of CXR interpretation with COVID-19 microbiologic testing. To the extent that such software products were and may in the future be used for this purpose, it makes sense to critically appraise their performance. However, the project of diagnosing COVID-19 via chest x-ray, whether by a human or by a computer, seems fundamentally misguided. As we’ve learned since 2020, clinically significant disease caused by SARS-CoV-2 can be consistent with a wide array of CXR findings including normal studies. Additionally, PCR positivity for SARS-CoV-2 can be consistent with acute disease, asymptomatic infection, and resolved past infection. Because of this underlying biology, it is difficult to know what to make of concordance or lack of concordance between CXR classification and microbiologic test results. While in the abstract there is a need to understand how radiology interpretation software of this type performs in the clinic, this study is hampered in that project by the fundamental unsuitability of any method of CXR interpretation to make the diagnosis of COVID-19.

The retrospective nature of the data collection also makes interpretation challenging. The study population included adults who presented to one of seven sites “who had symptoms consistent with COVID-19 based on a clinical evaluation” and for whom both a CXR and microbiologic test for SARS-CoV-2 was available. However, the symptoms assessed are not clear and Table 1 records that 85 of the included patients had either no symptoms or no record of symptoms. Also unclear is how both CXR and microbiologic testing were deployed. Over the course of the study period, many physicians around the world altered how they assessed COVID-19 in response to changing availability of resources, an evolving understanding of the disease, and alterations in patient presentation due to prior exposures and viral variation. It is clear that the subject population included in this study is heterogeneous. Per Table 1, at three sites 100% of the contributed cases were positive for COVID-19 by microbiologic testing. At one site, only 17% of contributed cases were positive. This suggests quite different underlying patient populations or, more likely, very different functional inclusion criteria due to practice patterns around symptom assessment and testing. While all included cases were assessed by both software models and radiologists, distinct groups of radiologists were used for cases from Zambia and Malawi. In particular for the analyses stratified by country, these practice pattern or patient population differences would be expected to affect some results of the study.

Several more minor points:

• 40 cases were excluded from the study because images could not be processed by the software used. This factor seems relevant to software performance and more detail about this limitation would be interested.

• The antigen and PCR tests for SARS-CoV-2 have very different performance characteristics as is obvious from Table 1, and concordance of CXR abnormalities with positivity by these two modalities would be expected to differ as well. Stratifying the analysis by type of COVID-19 test would substantially strengthen the manuscript.

• Software performance is discussed as being “non inferior” to the radiologist standard. A noninferiority margin should be explicitly defined and defended.

• The analysis stratified by time (Figure 4) is discussed in terms of the circulating viral strain being responsible for differences in software performance. But this is not justified. More than the viral strain differed over time during the pandemic, including patient triage practices, test availability, vaccination rates, and rates of prior exposure to the virus. 

• Several variables would be expected to interact to affect the sensitivity/specificity of COVID-19 diagnosis by CXR including the immune experience of the patient population over time, clinical practice as discussed above, viral strain, and COVID-19 test modality. However, it is never clear how these factors interact for any particular analysis. For example, was the Omicron strain equally represented in cases from Zambia and Malawi? Was PCR testing used with equal frequency across the analysis categories? 

• I am unclear on the justification for using distinct radiologist groups as the reference standard for Zambia and Malawi cases. This decreases power for the stratified analysis and introduces another potential bias to the pooled analyses. A common radiologist reference group would surely be preferable.

In sum, I think the authors make a convincing case that the two software products tested performed differently. However, conclusions beyond this are difficult to assess with the analyses presented.

Reviewer #2: This is a very well written study comparing the performance of different AI-guided tools to diagnose COVID-19 by chest x-ray with an experienced radiologist. The statistical methods were strong and the results were overall clearly communicated. Some important limitations should be addressed before recommending publication. 

Description is lacking in terms of the reference diagnostic process. Nearly all of the sample population was diagnosed with COVID-19 by nasal swab or nasopharyngeal swab, which can capture a wide range of presentations due to COVID-19 including asymptomatic disease, mild upper respiratory illness, and lower respiratory tract disease in addition to shedding without the disease. Three of those four categories would have no chest x-ray findings at all. It's helpful that the analysis limited comparisons to the same ground truth. If the data is available, then inclusion of the range of clinical criteria prompted microbiologic testing or a distribution of the population's presenting symptoms would strengthen the foundation of the study significantly to better understand the prior probabilities of lower respiratory tract COVID-19.

It's very helpful to note the lack of transparency regarding how these two models were trained. Is information such as the type of model or year that the models were published available? Such as a brief description broadly on whether these were convolutional neural networks, computer vision APIs, or something else. 

Would define the DICOM and FIND abbreviations prior to usage. 

Would clarify usage of the term "secondary analysis" which in one location describes it as evaluating the CXR-CAD for agreement with the radiologist and in another location uses the term to describe stratification by strain. 

Figure 1 is not immediately intuitive to communicate that participants who were both included and excluded are listed on the same diagram. A flow chart may be more clear, but otherwise a revised figure caption would also be helpful. Something like "the distribution of participants based on available data" would suffice, written prior to defining the categories as you have done. 

Would be careful of language stating "non inferior" unless a non-inferiority trial is being conducted, in which case a description of the margin for non-inferiority with statistical justification would be required. 

Where it is written "CAD2, which demonstrated a non-significant trend towards better overall performance, had an observed AUC...," it isn't clear to me which test is being referenced. Is this referring to the overall performance in figure 1 or specifically to figure 4b after listing the results of CAD2 in 4a? If the testing being referenced was a two-hypothesis statistical test, then it can't be certain if there is a trend or not. 

Otherwise there are several editorial notes that should be addressed. In many locations there are erroneously two spaces between words rather than one, a space between the period and the reference, a space before a period, inconsistent hyphenation of CXR-CAD or CXR CAD, and inconsistent format of the term SARS-CoV-2 and COVID-19. The figure and table captions are inconsistently formatted related to capitalization, spacing, italicization, in addition to the references in the text. The word "includes" in the table 1 caption and in the second line of the "Software interpretation..." section seems like it should say "included" and for figure 4 the abbreviations should be explicitly defined in the caption (they are listed in long form).

Otherwise the study is innovative, applicable for a highly relevant population, and has rigorous and focused statistical methods with acknowledged limitations. I look forward to the response.

6. PLOS authors have the option to publish the peer review history of their article (what does this mean?). If published, this will include your full peer review and any attached files.

**Do you want your identity to be public for this peer review?** For information about this choice, including consent withdrawal, please see our Privacy Policy.

Reviewer #1: No

Reviewer #2: Yes: Catherine Bielick

---

## [Decision Letter · Decision Letter 1]

7 Nov 2024

A Comparison of CXR-CAD Software to Radiologists in Identifying COVID-19 in Individuals Evaluated for Sars CoV-2 Infection in Malawi and Zambia

PDIG-D-24-00202R1

Dear Dr. Arrentz,

We are pleased to inform you that your manuscript 'A Comparison of CXR-CAD Software to Radiologists in Identifying COVID-19 in Individuals Evaluated for Sars CoV-2 Infection in Malawi and Zambia' has been provisionally accepted for publication in PLOS Digital Health.

Best regards,

Catherine G Bielick

Guest Editor

PLOS Digital Health

**Additional Editor Comments (if provided):**

**Reviewer Comments (if any, and for reference):**

Reviewer's Responses to Questions

**Comments to the Author**

1. If the authors have adequately addressed your comments raised in a previous round of review and you feel that this manuscript is now acceptable for publication, you may indicate that here to bypass the “Comments to the Author” section, enter your conflict of interest statement in the “Confidential to Editor” section, and submit your "Accept" recommendation.

Reviewer #2: All comments have been addressed

2. Does this manuscript meet PLOS Digital Health’s publication criteria? Is the manuscript technically sound, and do the data support the conclusions? The manuscript must describe methodologically and ethically rigorous research with conclusions that are appropriately drawn based on the data presented.

Reviewer #2: Yes

3. Has the statistical analysis been performed appropriately and rigorously?

Reviewer #2: Yes

4. Have the authors made all data underlying the findings in their manuscript fully available (please refer to the Data Availability Statement at the start of the manuscript PDF file)?

Reviewer #2: Yes

5. Is the manuscript presented in an intelligible fashion and written in standard English?

Reviewer #2: Yes

6. Review Comments to the Author

Reviewer #2: (No Response)

7. PLOS authors have the option to publish the peer review history of their article (what does this mean?). If published, this will include your full peer review and any attached files.

**Do you want your identity to be public for this peer review?** For information about this choice, including consent withdrawal, please see our Privacy Policy.

Reviewer #2: **Yes: **Catherine Bielick
